# Mechanisms of Action of Carbapenem Resistance

**DOI:** 10.3390/antibiotics11030421

**Published:** 2022-03-21

**Authors:** Caterina Aurilio, Pasquale Sansone, Manlio Barbarisi, Vincenzo Pota, Luca Gregorio Giaccari, Francesco Coppolino, Alfonso Barbarisi, Maria Beatrice Passavanti, Maria Caterina Pace

**Affiliations:** 1Department of Women, Child, General and Specialistic Surgery, University of Campania “L. Vanvitelli”, 80138 Naples, Italy; pasquale.sansone@unicampania.it (P.S.); vincenzo.pota@unicampania.it (V.P.); lucagregorio.giaccari@gmail.com (L.G.G.); francesco.coppolino@unicampania.it (F.C.); mariabeatrice.passavanti@unicampania.it (M.B.P.); mariacaterina.pace@unicampania.it (M.C.P.); 2Department of Medical, Surgical, Neurological, Metabolic and Aging Sciences, University of Campania “L. Vanvitelli”, 80138 Naples, Italy; manlio.barbarisi@unicampania.it; 3“Pegaso” Telematic University, 80143 Naples, Italy; alfonso.barbarisi@unicampania.it

**Keywords:** β-lactams antibiotics, carbapenem antimicrobials, antibiotic resistance, infections

## Abstract

Carbapenem antibiotics are the most effective antimicrobials for the treatment of infections caused by the most resistant bacteria. They belong to the category of β-lactams that include the penicillins, cephalosporins, monobactams and carbapenems. This class of antimicrobials has a broader spectrum of activity than most other beta-lactams antibiotics and are the most effective against Gram-positive and Gram-negative bacteria. All β-lactams antibiotics have a similar molecular structure: the carbapenems together with the β-lactams. This combination gives an extraordinary stability to the molecule against the enzymes inactivating the β-lactams. They are safe to use and therefore widespread use in many countries has given rise to carbapenem resistance which is a major global public health problem. The carbapenem resistance in some species is intrinsic and consists of the capacity to resist the action of antibiotics with several mechanisms: for the absence of a specific target, or an intrinsic difference in the composition of cytoplasmatic membrane or the inability to cross the outer membrane. In addition to intrinsic resistance, bacteria can develop resistance to antibiotics with several mechanisms that can be gathered in three main groups. The first group includes antibiotics with poor penetration into the outer membrane of bacterium or antibiotic efflux. The second includes bacteria that modify the target of the antibiotics through genetic mutations or post-translational modification of the target. The third includes bacteria that act with enzyme-catalyzed modification and this is due to the production of beta-lactamases, that are able to inactivate carbapenems and so called carbapenemases. In this review, we focus on the mode of action of carbapenem and the mechanisms of carbapenem resistance.

## 1. Introduction

Carbapenem antibiotics are considered the first line of treatment for infections caused by the most resistant bacteria such as: *Acinetobacter baumannii, Pseudomonas aeruginosa, Enterobacteriaceae,* (*Klebsiella pneumonia, Escherichia coli, Enterobacter* spp., *Serratia* spp., *Proteus* spp.,), *Enterococcus faecium, Streptococcus pneumoniae, Haemophilus influenzae, Shigella* spp. and *Mycobacterium tuberculosis.* The worldwide development of antibiotic resistance is considered a serious healthcare problem. In 2017, the World Health Organization (WHO) has given global priority to guide the research and to discover new antibiotics for an issued list of 12 antibiotic-resistant bacteria and particularly for carbapenem-resistant Acinetobacter baumannii (CRAB). These bacteria are part of the ESKAPE group (*Enterococcus faecium, Staphylococcus aureus, Klebsiella pneumoniae, Acinetobacter baumannii, Pseudomonas aeruginosa, Enterobacter species*) which approximately causes more than 2 million people in the United States to be infected with antibiotic-resistant bacteria annually, with 23,000 deaths as a direct result 4,5 [1,2,3]. Carbapenems are beta-lactams antibiotics such as other antibiotic penicillins (cephalosporins and monobactams). They are the most used antibiotics worldwide. All β-lactams antibiotics have a similar molecular structure: a beta-lactam ring [4,5,6]. Carbapenem have a penicillin-like five-membered ring, but the sulfur at C-1 in the five-membered ring is replaced by a carbon atom and a double bond between C-2 and C-3 has been introduced [5]. For carbapenems, the characteristic setting of the side chain in the trans position instead of the cis position, commonly found in other β-lactams, made them insensitive to the effects of β-lactamases. The PBPs 1a, 1b 2 and 3 are the principal targets of inhibition and PBPs 2 and 3 are specific for Gram-negative bacteria. Beta-lactam antibiotics can kill or even inhibit susceptible bacteria. The concentration of an antibiotic at the site of infection must be sufficient to inhibit the growth of the offending microorganism. If host defenses are impaired, antibiotic-mediated killing may be required to eradicate the infection. The concentration of the drug at the site of infection must not only inhibit the organism, but must also remain below the level that is toxic to human cells. If this can be achieved, the microorganism is considered to be susceptible to the antibiotic. The cell walls of bacteria are essential for their normal growth and development. Peptidoglycan is a heteropolymeric component of the cell wall that provides rigid mechanical stability. Gram-positive bacteria have as many as 40 layers of peptidoglycan, in Gram-negative bacteria, there appears to be only one or two layers. The surface structure in Gram-negative bacteria is more complex and the inner membrane, which is analogous to the cytoplasmic membrane of Gram-positive bacteria, is covered by the outer membrane, lipopolysaccharide and capsule. The outer membrane functions are an impenetrable barrier for some antibiotics. Other pharmacological differences that characterize the two types of bacteria are membrane permeability, efflux pumps, a large amount of different β-lactamase and the transpeptidation reaction, that occurs outside the cell membrane [5]. It is the last step in peptidoglycan synthesis that is inhibited by the β-lactam antibiotics [7]. Carbapenems are most effective against many bacterial infections such as *salmonellae bacteria, E.Coli, Mycobacterium tuberculosis, Methicillin-resistant Staphilococcus aureus (MRSA), Clostridium difficile, Klebsiella pneumonia* and *Pseudomanas aeruginosa.* They are safe to use and thus they have spread rapidly through all continents (many countries) giving rise to the carbapenem resistance [8]. The carbapenem resistance in some species is intrinsic such as *Staphylococcus aureus*, *Escherichia coli* and *Pseudomonas aeruginosa* and consists of the capacity to resist action of antibiotics with several mechanisms: for the absence of a specific target, or an intrinsic difference in the composition of cytoplasmatic membrane or the inability to cross the outer membrane. In addition to intrinsic resistance, bacteria can develop resistance to antibiotics with several mechanisms that can be assembled in three main groups. The first group includes antibiotics with poor penetration into the outer membrane of bacterium or antibiotic efflux. The second includes bacteria that modify the target of the antibiotics through genetic mutations or post-translational modification of the target. The third includes bacteria that act with enzyme-catalyzed modification. 

## 2. Mechanism of Resistance

### 2.1. Decreased Permeability

Resistance to β-lactam drugs may be related to the inability of antibiotics to reach their sites of action by reducing the entry into their outer cell walls through the porin channels. Some small hydrophilic antibiotics such as β-lactams, as well as tetracycline, chloramphenicol and fluorochinolones diffuse through aqueous channels in the outer membrane that are formed by proteins (Omp) called porins [9]. The number, the form and quality of porins in the outer membrane are mutable among different Gram-negative bacteria. A significant example is *P. aeruginosa* which is intrinsically resistant to a wide variety of antibiotics due its reduced classic high-permeability porins’ expression. Broader spectrum of β-lactam antibiotics such as ampicillin and amoxicillin and most of cephalosporins can diffuse through the pores in the *E. coli* outer membrane more rapidly than penicillin G. These are important instances of bacterial resistance to β-lactam antibiotics [10,11].

### 2.2. Overexpression of Efflux Pump

Active efflux pumps serve as another mechanism of resistance, removing the antibiotic from its site of action before it can act. This is an important mechanism of β-lactam resistance in *P. aeruginosa, *E. coli* and Neisseria gonorrhoeae*. Multidrug efflux pumps traverse both the inner and outer membranes of Gram-negative bacteria. The pumps are composed of a minimum of three proteins and are energized by the proton motive force and is produced by the respiratory enzymes and oxidative phosphorylation. Increased expression of these pumps is an important cause of antibiotic resistance. The onset of the mechanism of antibiotic resistance may represent a genetic occurrence [12,13,14]. The mutation responsible for drug resistance is usually a modification at a specific site on bacterial chromosomes. The capture, accumulation and dissemination of resistance genes are largely due to the actions of mobile genetic elements (MGE), a term used to refer to elements that promote intracellular DNA mobility (e.g., from the chromosome to a plasmid or between plasmids) as well as those that enable intercellular DNA mobility [15]. Thus, the resistance can be transferred from a resistant microorganism to a new location in the same or different DNA to a sensitive microorganism [16]. Multidrug efflux pumps are incorporated into bacteria and their purpose is to transport antibiotics outside the outer membrane of bacteria. Moreover, drug-specific efflux mechanism is promoted by plasmids and others mobile transporters [17]. All these components participate in specific activity promoting horizontal genetic exchange and contribute to achieve and disseminate the resistance genes. Bacteria efflux proteins are proteins identified primarily in Gram-negative bacteria but also exist in Gram-positive bacteria. They are divided into seven families. The seven families included in the resistance-nodulation-division (RND) superfamily are: the heavy metal efflux (HME), the nodulation factor export family (NFE), the major facilitator (MF) superfamily, the SecDF protein-secretion accessory protein family, the hydrophobe/amphiphile efflux-2 family, the eukaryotic sterol homeostasis family and the hydrophobe/amphiphile efflux-3 family. The efflux pumps of the RND superfamily, such as AcrB of *E. coli* and MexB of *P. aeruginosa,* play a fundamental role in promoting multidrug resistance. These pumps are associated with outer membrane channel such as TolC of *E. coli* and OprM of *P. aeruginosa* belonging to the OMF family proteins, the AcrA of *E. coli* and MexA of *P. aeruginosa* that are included into the MFP family. These three groups of proteins are essential for drug efflux and the mere lack of one these give the entire system a non-functional status. All these elements are involved in the intercellular mechanism of antibiotic resistance. Efflux mechanism or to be more precise, genes encoding efflux pumps are not commonly transmitted by mobile genetic elements. Only a few plasmid-mediated efflux pumps have been described in recent years. The transcription of genes involved in the efflux pumps is checked by local regulators. The overexpression of efflux genes, that are at the basis of Gram-negative bacteria resistance, is regulated by the mutation mechanism. 

### 2.3. Mutation and Transformation in Antibiotic Target Structures

Another mechanism which can develop antibiotic resistance is represented by the mutation of the antibiotic target. The resistance to streptomycin, to quinolones, to rifampin and others antibiotic groups is caused by a series of mutations that can occur in the gene encoding the target protein, the protein concerned with drug transport and/or with drug activation. [18] The archetypal example of this is penicillin resistance in S pneumoniae and which is conferred by mosaic penicillin-binding protein (pbp) genes encoding penicillin-insensitive enzymes. Mosaicism in the penA gene, which encodes a PBP, in N.gonorrhoeae has also been linked with high-level resistance to extended-spectrum cephalosporins. Every antibiotic has its specific target site, by binding to which exerts its inhibitory effect. Mutation of the target site can occur during an infection and a mutation into a single microorganism can induce resistance [19]. Genes encoding DNA gyrase or topoisomerase IV, within the bacterial chromosomal are responsible for quinolone resistance, this resistance is expanded above all after the therapeutic use of these antibiotics in Pseudomonas and Staphylococci [20]. The mechanism of mutation may occur in a functional target with a decreasing affinity for the antibiotics, which do not work completely and behave with a reduced efficiency (Table 1). Another mechanism of bacterial resistance to antimicrobial agents is the transformation. This is the capacity of transferring genetic information, involving uptake and incorporation of DNA, how it happens in altered penicillin-binding proteins (PBPs), that are mosaics of overseas DNA imported and incorporated from a streptococcus to a resident PBP gene [21].

### 2.4. Modification of Antibiotics by the Hydrolysis of the Molecule

The modification of antibiotics by hydrolysis is a major mechanism of antibiotic resistance that has been relevant since the first use of antibiotics, with the discovery of penicillinase (a β-lactamase) in 1940. Afterwards, the β-lactam-hydrolyzing enzymes have broadened their activity from penicillinases, followed by cephalosporinases, then extended to spectrum β-lactamase ESBLs and most recently, to the mannose-binding lectine MBLs and other carbapenemases [22]. The MBLs is characterized by the ability to hydrolyze carbapenems (often considered as last resort drugs) and bacteria producing these β-lactamases including *A baumannii, K pneumoniae and E.coli* render many of the beta-lactam class of antibiotics such as penicillins, cephalosporins, monobactams and carbapenems ineffective [22]. Beta-lactamases are classified into four main groups based on their amino acid sequences (classes A, B, C and D). The production of all four classes of beta-lactamases (A, B, C and D) is generally chromosomally encoded [23,24]. Class A includes extended-spectrum β-lactamases (ESBLs) and Klebsiella pneumoniae carbapenemase enzymes; class B enzymes are the metallo- β-lactamases (MBLs) that have a broad substrate range, being able to inhibit all beta-lactam antibiotics except the monobactams [25]; class C enzymes are the cephalosporinases, broadly disseminated enzymes usually resistant to cephamycins (cefoxitin and cefotetan), penicillins and cephalosporins [26,27]; and class D enzymes are oxacillinases. Recently it has been recognized that *A.baumannii,* although often considered a less virulent pathogen compared with *K. pneumoniae and P.aeruginosa*, plays a significant role in spreading broad-spectrum resistance genes to other Gram-negative organisms [28].

The increase in ESBL-producing bacteria has enlarged the clinical use of carbapenems, as well as increasing the number of carbapenem hydrolyzing activity. These enzymes, known as carbapenemases, are a large variety and were identified for the first time in *Enterobacteriaceae* and were divided into the Ambler four classes of β-lactamases, class A, B, C and D, and all are characterized by having a part in common of serine in the target. Their function is to inactivate β-lactame antimicrobials and among these mostly carbapenems [29,30]. The hallmark of these enzymes is the ability to inactivate a broad range of β-lactams, including carbapenems and extended-spectrum cephalosporins. Although first identified on the chromosomes of single species, many carbapenemases are now plasmid-mediated and have been reported in Enterobacteriaceae, *P. aeruginosa* and *A. baumannii* [31]. The spread of carbapenemases has occurred in different ways, as exemplified by the *kpc* and *ndm* genes. The serine carbapenemase KPC was first identified in *K. pneumoniae* in 1996 [32] and has since been described within several species of Enterobacteriaceae [33]. The *kpc* gene is plasmid-borne and is associated with a dominant clone of KPC-producing *K. pneumoniae*, ST258, which is found worldwide [34]. This gene is often carried on the pKP-Qil plasmid or on closely related variants [35], and there are several variants of the *kpc* gene that encode proteins that can be differentiated by single amino acid substitutions (although most retain similar activity); KPC2- and KPC3-producing strains have been responsible for outbreaks in the United States, Greece, Israel and the United Kingdom [36,37]

#### 2.4.1. Class A Carbapenemases

Class A carbapenemases are chromosomally encoded (SME, NmcA, SFC-1, BIC-1, PenA, FPH-1, SHV-38), plasmid-encoded (KPC, GES, FRI-1), or both (IMI) [27]. Among these, the best known KPC (*Klebsiella pneumoniae carbapenemase*) was spread all around the world and has been isolated in most of the clinical enterobacterial species such as *P. aeruginosa and A. baumannii*. Generally, class A carbapenemases reduce susceptibility to imipenem for the bacteria sensitive to it and allow the hydrolysis of a broad variety of β-lactams, including carbapenems [38].

#### 2.4.2. Class B Carbapenemases

The class B of the Ambler group belonging to the (MBLs) are clinically the most relevant carbapenemases. These are divided into three subclasses: B1, B2, and B3, but the largest number of clinically relevant MBLs belong to the B1 subclass, including the most frequently Verona integron-encoded MBL (VIM), imipenemase (IMP) and New Delhi MBL. (NDM) Those MBLs are generally located within different integron structures, which are connected with mobile plasmids or transposons facilitating the transfer of resistance genes between bacteria [29].

#### 2.4.3. Class D Carbapenemases

Class D carbapenemases (Oxa-type β-lactamases, CHDLs) are a group of enzymes that have been discovered in A. baumannii and K. pneumoniae human strains [39]. These carbapenemases and specifically the OXA-48 and related variant are of clinical relevance because they make it difficult to treat infection [12]. In clinical therapy, the β-lactamase inhibitors (i.e., amoxicillin-clavulanic acid etc.) are effective for class A β-lactamases, but do not inhibit class D carbapenemases. Therefore, it is important in the next future to develop inhibitors of A. baumannii class D carbapenemases to make the carbapenem antimicrobials effective [29,40] (Table 2).

#### 2.4.4. Class C Carbapenemases

Class C enzymes are not considered carbapenemases. However, it has been shown that they possess a low potential of carbapenem hydrolysis, and their overproduction may contribute to carbapenem resistance combined with diminished outer-membrane permeability or efflux pump overexpression [41].

## 3. Discussion

Therapy with β-lactam antibiotics is dynamic. The prevalence of bacterial resistance to these agents continues to rise, while new and more effective agents are released for clinical use. The effort against infectious diseases is probably one of the greatest public health challenges faced by our society, especially with the emergence of carbapenem-resistant Gram-negative species that are in some cases pan-drug resistant [42]. β-lactam antibiotics are essential and the most prescribed antibiotics that share a common structure and mechanism of action: the inhibition of synthesis of the bacterial peptidoglycan cell wall. Although bacteria can develop resistance to antibiotics, β-lactam antibiotics cannot kill or even inhibit all bacteria and various mechanisms of bacterial resistance to these agents are operative. The bacterial resistance can be caused by the inability of the agent to penetrate to its site of action or by energy-dependent efflux systems for pumping the antibiotics out of the bacteria. Bacteria can also destroy β-lactam antibiotics enzymatically.

Different microorganisms elaborate a number of distinct β-lactamases, although most bacteria produce only one form of enzyme. In general, Gram-positive bacteria produce a large amount of β-lactamase which is secreted extracellularly. In Gram-negative bacteria, β-lactamases are found in relatively small amounts, but are located in the periplasmic space between the inner and the outer cell membranes. Since the enzymes of cell wall synthesis are on the outer surface of the inner membrane, these β-lactamases are strategically located for maximal of the microbe.

β-lactamases of Gram-negative bacteria are encoded either in chromosomes or in plasmids and these enzymes, together carbapenemases, may hydrolyze a broad variety of β-lactam antibiotics such as penicillins, cephalosporins, aztreonam and carbapenems. Carbapenems have the broadest antimicrobial spectrum of any antibiotic, the main problem is the bacterial resistance against β-lactam antibiotics that continues to increase at a dramatic rate. Many factors are involved, among these are the greater number of relatively resistant microorganisms in a large population, the amount of [43] β-lactamase produced and the phase of growth of the culture.

## 4. Conclusions

The production of carbapenemases is the major mechanism underlying carbapenem resistance around the world and represents a great health concern. More knowledge is needed to control resistant genes and resistant organisms and their dissemination. There is an urgent need for a global collaboration to plan valid strategies to prevent the spread of carbapenemases and the development of new antimicrobial molecules.

## Figures and Tables

**Table 1 antibiotics-11-00421-t001:** Molecular mechanisms of antibiotic resistance.

↓ Permeability	Outer membrane forms a permeability barrier (Gram positive >Gram negative). Down-regulation of porins or by the replacement of porins with more selective channels.
↑ Efflux	Bacterial efflux pumps actively transport many antibiotics out of the cell (multidrug resistance [MDR] efflux pumps).
Mutation and Transformation in Antibiotic Target Structures	Changes to the target structure that prevent efficient antibiotic binding: Transformation can confer antibiotic resistance by target protein modification through the formation of ‘mosaic’ genes. Acquisition of a gene homologous to the original target. Protection by modification of the target: Erythromycin ribosome methylase (erm). Chloramphenicol–florfenicol resistance (cfr) methyltransferase. Quinolone resistance (qnr) gene.

**Table 2 antibiotics-11-00421-t002:** Molecular classification of carbapenemase enzymes.

Class A	Chromosomally encoded—NmcA (not metalloenzyme carbapenemase A)	- SME (Serratia marcescens enzyme). - IMI-1 (Imipenem-hydrolysing β-lactamase). - SFC-1 (Serratia fonticola carbapenemase-1).
	Plasmid encoded	- KPC (Klebsiella pneumoniae carbapenemase, KPC-2 to KPC-13). IMI (IMI-1 to IMI-3). - GES (Guiana extended spectrum, GES-1 to GES-20).
Class B	β-lactamases inhibited by EDTA	- NDM-1 (New Delhi metallo-β-lactamase 1). - IMP (Imipenem-resistant Pseudomonas). - VIM (Verona integron-encoded metallo-β-lactamase). - GIM (German imipenemase). - SIM (Seoul imipenemase).
Class C	Emerging class resistant to penicillin, oxyiminocephalosporins, cephamycins (cefoxitin and cefotetan), and, variably, to aztreonam.	- ACT (AmpC type). - CMY (Cephamycinase). - ADC (Acinetobacter-derived cephalosporinase).
Class D	β-lactamases poorly inhibited by EDTA or clavulanic acid	- OXA (Oxacillin-hydrolyzing carbapenemases).

## Data Availability

The data presented in this study are available on request from the corresponding author.

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
