# Peer review of "Mechanisms of Action of Carbapenem Resistance"

_antibiotics, 2022, doi:10.3390/antibiotics11030421_

Round 1

Reviewer 1 Report

The manuscript entitled “MECHANISMS OF ACTION OF CARBAPENEM RESISTENCE” describes the mode of action and the mechanisms of resistance in particular of carbapenem resistance. This is a compact review. The following items should be revised in the submitted manuscript. The related references should be added to the revised manuscript with given information.

The language of the article should be revised. Attention should be paid to the spelling of the names of bacteria and the use of punctuation marks.

Comments:

Page 1, line 41: “Carbapenem antibiotics are considered the first line treatment for infections caused 40 by most resistant bacteria” most resistant bacteria is so general definition it should be specified.

Page 1, Line 42: “a great number of infections” it should be given in numbers it is so general. The number of infection cases caused by resistant bacteria.

Page 2, Line 47: change as “Pseudomonas aeruginosa, and Enterobacter spp.”

Page 2, Line 48: “causes more infections” does not make a sense. Please use numbers or percentages in scientific sounds.

Page 2, Line 57: “Beta-lactam antibiotics can kill or even inhibit susceptible bacteria.” Why they are susceptible could be explained here.

Page 2, Line 58: “for different basic phenomenon” does not make a sense.

Page 2, Line 59: “gram-positive and gram-negative cell walls is different” This can be removed. Please use Gram-positive and Gram-negative throughout the manuscript not gram.

Page 2, Line 67: it should be added a reference related cell wall structure of bacteria.

Page 2, Line 71: “many bacterial infections” please give examples of infections. Avoid to use many, more, much, and some definitions. Use numbers or percentages or comparable results.

Page 2, Line 72: “in many countries” give examples?

Page 2, Line 73: “in some species” give examples?

Page 2, Line 85: “Some small hydrophilic antibiotics” give examples of the antibiotics?

Page 2, Line 87: “…formed by proteins (Omp) called porins.” Give a related reference.

Page 3, Line 97: “P. aeruginosa, E. coli, and Neisseria gonorrhoeae” “and” should be non-italic.

Page 3, Line 101: add related reference

Page 3, Line 109-110: “These factors involved in the widespread of resistance are especially present between gram-negative bacteria, Here the formation” remove “here”

Page 3, Line 126: “the eavy” should be “heavy”

Page 3, Line 130: “AcrB of Escherichia coli”  change as E.coli write italic

Page 3, Line 131: “MexB of Pseudomonas aeruginosa” change as P. aeruginosa write italic

Page 3, Line 141: “many times regulated” does not make a sense.

Page 3, Line: 146-147: “The resistance to streptomycin, to quinolones, to rifampin 146 and others antibiotic structures is regulated” change as “The resistance to streptomycin, quinolones, rifampin, and other antibiotic groups like ….is regulated”

Page 3, Line 149: “Many proteins” give examples? Describe these proteins?

Page 4, Line 154: use “microorganisms” instead of “germs”

Page 4, Line 157: various pathogens? Describe or give examples?

Page 4, Line:160: “and Staphylococci” they are not italic

Page 4, Line:165 “penicillin-binding proteins (PBPs)” use only “PBPs”

Table 1.  change as (Gram-negative > Gram-positive).

Table 1. Mutation and transformation in antibiotic target structures. There are numbers 1. 2. 3. But the numbers given are not explanatory.

Page 4, Line 174-175: write the open name of abbreviations in the usage first time. ESBL?, MBL?

Page 4, Line 177: “multi-resistant Gram-negative bacilli” give examples?

Page 4, Line 177: give open name of GNB

Page 4, Line 178: change as “through plasmids that contain β-lactamases and they are…”

Page 4, Line 179-182: please re-write this sentence. “The production of all four classes of -lactamases (A, B, C, and D) through the incorporation of exogenous DNA into its genome would underlie the rapid evolution of this strain toward multi-resistance [21-22]. (GES-11 and CTX-M) [23].”

Page 4, Line 183: change as K. pneumoniae

Page 4, Line 188: change as A. baumannii

Page 4, Line 188: change as “Recently, it has been recognized that”

Page 5, Line 189-190: K. pneumoniae and P. aeruginosa

Page 5, Line 197: penicillin-binding proteins (PBPs) use only “PBPs”

Page 5, Line 200: use only ESBLs

Page 5, Line 202: These enzymes

Page 5, Line 206-207: use only ESBLs and K. pneumoniae

Page 5, Line 202-204: please check this information with Page 4 line 178-182. Avoid repetition.

Page 5, Line 223-225: change as “Recently it was recognized that A. baumannii, although often considered a less virulent pathogen compared with K. pneumoniae and P. aeruginosa, plays a significant role in spreading broad-spectrum resistance” and add related reference.

Page 5, Line 227-228: change as” Class A carbapenemases are chromosomally-encoded (SME, NmcA, SFC-1, BIC- 227 1, PenA, FPH-1, and SHV-38), plasmid-encoded (KPC, GES, and FRI-1), or (IMI).” Add related reference.

Page 5, Line 233: P. aeruginosa should be italic

Page 5, Line 235: use only MBLs

Page 5, Line 237: add related reference

Page 5, Line 239: use only MBLs

Page 6, Line 244: add related reference.

Page 6, Line 248-249: change as “that have been discovered in A. baumannii and K. pneumoniae human strains.” Add related reference.

Table 2. Do not write the names of the enzymes with italic.

Page 6, Line: 266: add related reference.

In reference section the names of bacteria should be written italic. Like ref 3, Pseudomonas aeruginosa. Please check again all references.

Author Response

Dear reviewer thanks  for your review. I've done a ponti-by-point correction of all your queries. Please see the attachment.

Reviewer 2 Report

The article is a very general overview. I understand that Narrative Review may not be a complete literature review and not be a particularly in-depth description of the topic, but it should be guided by a keynote, be  interesting to the reader. I'm sorry to say this, but it's also written wrong. I have the impression that each subsection was written by a different person. With all due respect, the Authors lack precision, good translation or lack of knowledge of molecular biology. Free expression must not lead to a loss of scientific presentation of the facts. The entire article needs to be redrafted.

Abstract- Please rewrite the abstract. Shorten long sentences and correct mistakes.

Line 21-23: This sentence must be corrected.

Line 23-24: “ For these reasons carbapenems are most effective against many bacterial infections, mainly among Gram-negative bacteria” - For what reasons? Maybe it is the mental shortcut. The structure of the antibiotic molecule itself is not responsible for its effectiveness against infection. This is also influenced by pharmacokinetics. Please correct.

Line 36: Why "in particular"? - it will be about other antibiotics?

Line 41-42: ”The worldwide developing of antibiotic resistance has performed the one of the principal choice for a great number of infections” - The sentence has no sense.

Line 48: “causes more infections with antibiotic resistance” - The mental shortcut. Infections can be caused by antibiotic-resistant bacteria.

Line 48-50: Rewrite the sentence.

Line 50-53: “Beta-lactams antibiotics have in common a similar molecular structure: the carbapenem together with the β-lactam ring” - That's not true. The common structure is based on the beta-lactam ring and carbapenems have a penicillin-like five-membered ring, but the sulfur at C-1 in the five-membered ring is replaced with a carbon atom and a double bond between C-2 and C-3 is introduced.

Line 52- 54: For carbapenems the characteristic setting of the side chain in the trans position instead of the cis position commonly found in other β-lactams, made them insensitive to the effects of β-lactamases .

Line  55-56:  This sentence structure changes the meaning. PBP proteins are not involved in inhibition, they are the target of inhibition.

Line 57-58: “Gram-negative bacteria are more resistant than Gram-positive bacteria for different basic phenomenon”  - I have no idea what the Authors were trying to say.

Line 59:  Please standardize the spelling: Gram or gram.

Line 62-63: Wrong spelling or abbreviation. We can use the phrase "molecule" to peptidoglycan when we think about  cross-linked chains, as each peptidoglycan layer is a single giant molecule. But here we use  " Gram-positive bacteria, has  as many as 40 layers of peptidoglycan, in Gram-negative bacteria, there appears to be only one or two layers.

Line 71:  “What all the reasons ?? no logical sequence.

Line 72: “safe to use “ -  Is it safe due to the structure of the cell wall described above? mental shortcut, no logical sequence.

Line 104: “due to the inhibition of mobile genetic elements (MGE)” -  What does it mean? this part of the text doesn't make sense.

Line 105: "This resistance:"- what are the authors referring to? to efflux pumps?  Most of the efflux pumps are located on the chromosome of bacteria.  No logical sequence in the chapter.

Line 108- 109:  Insertion sequences are the part of integrons, integrons can be the part of transposons. We present data  in a particular order. A better idea is to transfer all information about mobile genetic elements to the Introduction.

Line 110:  Why does the description of efflux pumps mention the hydrolysis mechanism of the antibiotic molecule? besides this is too long sentence, please shorten it.

Line 113: ” Multidrug efflux pumps are incor-113 porate into bacteria”-  The mental shortcut, please write precisely.

Line 115- 117: Repetition of the information already provided.

Line 117- 120: The sentence is too long, makes no sense.

Line 122- 124: The sentence is grammatically incorrect, it loses its sense.

Line  129: Repetition

Line 136: Why is an unrelated article [15] (about RNA methyltransferase) cited here?

Line 136- 137: Efflux mechanism or to be more precise, genes encoding efflux pumps are not commonly transmitted by mobile genetic elements . Only a few plasmid-mediated efflux pumps have been described in recent years, such as QacBIII, Tet(L) or MexCD efflux pumps.

Line 137- 143: Please change these sentences. This is rather a general repetition. The entire subsection needs to be edited.

Line 146-147: Rather "is caused by" not "regulated" by mutations.  The sentence is too long.

Line 149-151: I have no idea what the Authors were trying to say. The literature cited here is  "Lipopolysaccharide-Deficient Acinetobacter baumannii Due to Colistin Resistance Is Killed by Neutrophil-Produced Lysozyme". LPS loss is not the mechanism based on mutation of colistin target. Colistin is a surface active agent acts by "detergentlike" mechanism.

Line 153-155: Terrible sentence. Please correct.

Line 155-160:  Why are the authors describing the basics of resistance mechanisms to other antibiotics, rather than focusing on PBP proteins?

Line 170: Wrong grammatical structure of the sentence. Proper sense “Modification of antibiotics by the hydrolysis of the molecule” or just “the hydrolysis of an antibiotic”.

Line 171: Wrong grammatical structure of the sentence.

Line 175: “The MBLs have mainly affected the necessity of carbapenems (often considered as last resort drugs) which are used for the management of multi-resistant Gram-negative bacilli”  -  The sentence doesn't make sense. How  MBLs – metalo-beta-lactamases (carbapenemases) affected the necessity of carbapenems??

Line 177-179: The sentence is too long, makes no sense.  “GNB”- What does this abbreviation mean? It only shows up here.

Line 179-180: „The production of all four classes of -lactamases (A, B, C, and D) through the incorporation of exogenous  DNA into its genome” The sentence has no sense- production through the incorporation of exogenous  DNA?

Line 180: “[21-22]. (GES-11 and CTX-M) [23].” ???

Line 190-191: “contain the genes encoding for narrow-spectrum-lactamases (i.e., TEM-1, SCO-190 1, and CARB-4) and those encoding for ESBL that play a significant role in spreading 191 broad-spectrum resistance.”  What is the point of this sentence? ESBLs themselves do not play a role in the spread of resistance (only mobile elements carrying the genes encoding them). Similar errors (inaccuracy) are found throughout the article.

Line 192- 194: Why describe the different classes of carbapenemases in (only) A. baumanii here?  

Why is the article [26] not related to A. baumanii cited here? (Safety of Meropenem in Patients Reporting Penicillin Allergy: Lack of Allergic Cross Reactions) ?

Line 194-200: This is the repetition from the Introduction. It is completely unnecessary here!

Line 206-210: Another repetition in the same subsection.

Line 214: “The spread of carbapenemases has occurred in different ways” - Carbapenemases are not able to spread- only the genes that code for them can be spread. Such abbreviations (mental shortcut) cannot be used in a serious scientific article .

Line 216: ” described in several Enterobacteriaceae”  it should be written - within several species of Enterobacteriaceae.

Line  223- 224: “that Acinetobacter baumannii, although often considered a less virulent pathogen compared with K pneumoniae and Pseudomonas aeruginosa”.  Repetition. Exactly the same words are at the top of the page.  

Line  226: The characterization of class carbapenemases should be in subsection 1.4.1, and so on.

Line 228: Pleonasm? Plasmid-encoded enzymes are encoded on plasmids. This is probably obvious. Plasmids are mobile elements. Plasmid-encoded enzymes can't be encoded on chromosomes. This is another example of imprecision, poor translation or lack of knowledge of molecular biology.

Line 233-235:  Another repetition. The section on class A carbapenemases should describe class A carbapenemases. The classification has already been presented.

Line 249-252: Badly worded sentence. The subject in the sentence is "carbapenemases" so this sentence indicates that carbapenemases- enzymes- are responsible for majority of A. baumanii infections. This is not true. The virulence factors of the bacteria are mostly responsible for the infections. The production of carbapenemases by the bacteria makes the infection difficult to treat. 

Line 253-254: “but does not in-hibit class D carbapenemases.” Avibactam inhibit some class D enzymes such as OXA-48 from Klebsiella pneumoniae, and OXA-24, OXA-40 and OXA-69 from Acinetobacter baumannii.

1.8. Class C Carbapenemases- A subsection consisting of one long sentence….

Line 272: “gram-negatives”  - Gram-negative species.

Line 272-274: Two thoughts in one sentence.

Line 275-277: Wrong grammatical structure of the sentence.

Line 278: “be present in large quantities”- The mental shortcut, please expand.

Line  285: Why is an article [46] cited here that does not focus on beta-lactams? The source of data for the Article should be literature focused on the subject, thoroughly examining the subject discussed.

Line 289: “The main problem related to carbapenems continues to be the clinical treatment of serious infections with antibiotics whose activity is compromised by the production of carbapenemase and the epidemiological spread in most bacteria in the main geographic areas.”  Poorly constructed, long sentence in which one don't know what the subject is.

Line 292.  In this place other article of the most Authors of the Manuscript is cited . Self- citations is not a mistake as long as concerns a closely discussed topic. In the Reference 47 the case of septic shock due to Escherichia coli meningoencephalitis is described. The treatment was not associated with an antibiotic resistant strain, and no mention was made of antibiotic resistance at all. 

Author Response

Dear Reviewer I've done all the correction you have required and rewrite some part of it. Please see the attachment.

Round 2

Reviewer 1 Report

The required modifications were done by the authors. It can be accepted for publication.